# Calcifediol (25OHD) Deficiency and Its Treatment in Women’s Health and Fertility

**DOI:** 10.3390/nu14091820

**Published:** 2022-04-27

**Authors:** Ana Arnanz, Juan A. Garcia-Velasco, José Luis Neyro

**Affiliations:** 1IVIRMA, 28023 Madrid, Spain or ana.arnanz@edu.uah.es (A.A.); juan.garcia.velasco@ivirma.com (J.A.G.-V.); 2Departamento de Biomedicina y Biotecnología, Universidad de Alcalá de Henares, 28023 Madrid, Spain; 3Academia de Ciencias Médicas de Bilbao, Gynecology and Obstetrics Service, Hospital Universitario Cruces, 48009 Bilbao, Spain

**Keywords:** calcifediol, vitamin D deficiency, women’s health, fertility

## Abstract

Currently, there is abundant scientific evidence showing that the vitamin D endocrine system (VDES) is a highly complex endocrine system with multiple actions in different regions of the body. The unequivocal presence of vitamin D receptors in different tissues related to fertility, and to specific aspects of women’s health such as pregnancy, undoubtedly implies functions of this steroid hormone in both male and female fertility and establishes relationships with different outcomes of human gestation. In order to review the role of the VDES in human fertility, we evaluated the relationships established between 25-hydroxyvitamin D (calcifediol) deficiency and in vitro fertilization, as well as aspects related to ovarian reserve and fertility, and commonly diagnosed endocrinopathies such as polycystic ovary disease. Likewise, we briefly reviewed the relationships between calcifediol deficiency and uterine fibroids, as well as the role that treatment may have in improving human fertility. Finally, the best scientific evidence available on the consequences of calcifediol deficiency during pregnancy is reviewed in relation to those aspects that have accumulated the most scientific literature to date, such as the relationship with the weight of the newborn at the time of delivery, the appearance of preeclampsia, and the risk of developing gestational diabetes and its final consequences for the pregnancy. To date, there is no definitive consensus on the necessary dose for treatment of calcifediol deficiency in the therapeutic management of infertility or during pregnancy. Large prospective clinical intervention studies are needed to clarify the benefits associated with this supplementation and the optimal dose to use in each situation. Although most intervention studies to date have been conducted with cholecalciferol, due to its much longer history of use in daily care, the use of calcifediol to alleviate 25-hydroxyvitamin D deficiency seems safe, even during pregnancy. The unequivocal presence of vitamin D receptors in very different tissues related to human fertility, both male and female, as well as in structures typical of pregnancy, allows us to investigate the crucial role that this steroid hormone has in specific aspects of women’s health, such as pregnancy and the ability to conceive. Well-designed clinical studies are needed to elucidate the necessary dose and the best form of treatment to resolve the very common calcifediol deficiency in women of reproductive age.

## 1. Introduction

When cholecalciferol was described a century ago, it was called vitamin D because it was erroneously thought to be a vitamin, and it was called D because it was the fourth vitamin described. However, we now know that it is not a vitamin; rather, it is a threshold nutrient which is part of an endocrine system, the highly structural and functionally complex vitamin D endocrine system (VDES) [1], and it is similar to other steroid hormones.

Calcifediol, which is produced by the action of hepatic 25-hydroxylase (CYP2R1/CYP27A1), is the prohormone and cornerstone of the VDES and the substrate for synthesizing 1,25-dihydroxyvitamin D3(1,25(OH)2D (calcitriol), the active hormonal form of the VDES, via the action of 25(OH)D-1α-hydroxylase (CYP2721B) in the kidneys (controlled endocrinologically); in many other cells of the body, the process is under autocrine/paracrine control [1].

The VDES has receptors (VDRs) belonging to the superfamily of nuclear steroid receptors that use the same heterodimer partner (RXR) on virtually all cells in the human body [1], including in the hypothalamus, pituitary gland, ovary, uterus, thyroid, testis, and prostate, as well as the placenta, yolk sac, and embryonic muscle [2,3].

Calcitriol-activated VDRs (VDES/VDR) regulate the transcription of ~3% of genes, with a broad spectrum of functional activities which determine the systemic and auto/paracrine endocrine action of the VDES; in addition, calcitriol can exert other rapid, nongenomic actions, which occur within seconds to minutes after hormone binding, mediated by a membrane-bound form of the VDR [1]. The VDES promotes calcium absorption in the gut and maintains adequate serum calcium and phosphate concentrations to enable normal bone metabolism. In addition to bone and calcium metabolism, the VDES has polymorphic roles in the body, regulating multiple physiological processes in other organs and systems (e.g., neuromuscular, cardiovascular, and the innate and adaptive immune systems), and regulates cell growth and hormone secretion, glucose metabolism, xenobiotic metabolism, and inflammation. The presence of VDRs in sites related to fertility (granulosa cells, endometrium) and pregnancy unequivocally confirms the necessary role of this steroid hormone in the development of both male and female fertility as well as in gestational outcomes [4,5].

It is well described that the most important source of vitamin D for humans is exposure of the skin to sunlight (80–90%), with less than 10–20% being derived from the diet [6,7]. However, the direct measurement of circulating vitamin D itself is not a good marker of the nutritional status of the VDES. Immediately after its cutaneous synthesis or intestinal absorption, it rapidly disappears from circulation. Thereafter, it reappears as 25(OH)D (calcifediol), tightly bound to vitamin D binding protein, which has a long half-life (2–3 weeks) and a higher concentration and is also the essential substrate for the synthesis of 1,25(OH)2D, or calcitriol, the VDES hormone [1,7].

These data have generated a universal consensus that the measurement of total circulating 25(OH)D concentration constitutes a robust and reliable biomarker of the nutritional status of the VDES [1,8]; thus, the quantification which is currently called “vitamin D measurement”, should be relabeled 25(OH)D, or calcifediol, measurement. The measurement of circulating 25(OH)D, or calcifediol, is used by health authorities and scientific societies in America and Europe to establish normality status, the definition of “vitamin D” deficiency and degree of “vitamin D” insufficiency used to establish dietary reference intakes, and the values for “vitamin D”, as well as population monitoring of “vitamin D” deficiency, insufficiency, or excess, or, more appropriately, calcifediol deficiency, insufficiency or excess.

Oral treatment with calcifediol (25OHD3) itself, rather than supplementation with vitamin D, should also be considered for the correction of its deficiency or insufficiency [6]. Oral calcifediol has a higher rate of intestinal absorption and a linear dose–response curve irrespective of baseline serum levels of 25(OH)D, and intermittent intake of calcifediol results in fairly stable serum 25(OH)D compared with greater fluctuations after intermittent oral cholecalciferol. All of the above make calcifediol more than three times more potent than cholecalciferol [6].

The objective of this manuscript is to review the influence that the VDES has on human fertility, as well as its presence in the diseases that most commonly complicate pregnancy, and to observe the final results of pregnancy based on serum levels of 25(OH)D/calcifediol, according to the available literature.

## 2. The Vitamin D Endocrine System and Infertility

According to the most recent international guidelines, the VDES nutritional status of a patient is defined as deficient when 25(OH)D/calcifediol serum levels are below 20 ng/mL, insufficient when 21–29 ng/mL, and replete if above 30 ng/mL [9].

The VDES appears to play an important role in the physiology of the female reproductive system; however, one of the biggest challenges is in understanding whether the VDES has a potential influence on folliculogenesis and oogenesis, on endometrial receptivity, or on both.

A recent meta-analysis reported a high prevalence of 25(OH)D/calcifediol deficiency among women undergoing in vitro fertilization (IVF) treatment; of these women, 33.7% were classified as deficient, 38.5% as insufficient, and only 27.8% as sufficient [10]. The available evidence regarding the role of 25(OH)D in assisted reproduction remains conflicting.

### 2.1. VDES/VDR and Embryo Development/Implantations/Clinical Pregnancy/Live Birth Rate after In Vitro Fertilization/Intracytoplasmic Sperm Injection (ICSI)

As previously shown in animal studies, VDES/VDR may affect embryogenesis and follicle development [11]. The effect of the VDES on folliculogenesis and IVF treatment outcomes has been studied by measuring 25(OH)D/calcifediol levels not only in serum but also in follicular fluid, as they are positively correlated [12]. Evaluation of 25(OH)D values at the intrafollicular level might improve our understanding of the mechanism of action by which 25(OH)D/calcifediol may affect oocyte/embryo competence in IVF, since the follicular biochemical environment might directly affect the quality of the oocyte [13]. There is clinical evidence that lower 25(OH)D levels in follicular fluid are associated with lower embryo quality, fertilization, implantation, and clinical pregnancy rates [14] (Table 1). In contrast, Ciepiela et al. showed that embryo quality is improved in patients with lower levels of 25(OH)D/calcifediol in pooled follicular fluid at the time of oocyte retrieval [15]. Furthermore, high 25(OH)D levels (>30 ng/mL) in follicular fluid also seem to negatively affect embryo quality and IVF outcomes [16]. An increased level of 25(OH)D/calcifediol in follicular fluid, in combination with a decrease in glucose levels, may have an adverse effect on embryo development, indicating a detrimental effect at the oocyte level [16].

Based on these observations, intrafollicular 25(OH)D/calcifediol might be a marker of oocyte/embryo competence at the chromosomal level. Interestingly, in a recent randomized controlled trial (RCT) in a Middle Eastern population, a positive correlation was found between VDES metabolite levels in individual follicles and blastocyst chromosomal status in patients undergoing IVF treatment, demonstrating that patients with an adequate level of 25(OH)D/calcifediol have a higher probability of having a euploid blastocyst than 25(OH)D/calcifediol-deficient individuals [17].

Most studies have failed to find any correlation between 25(OH)D/calcifediol status in serum and embryo morphological parameters [18,19]. No correlation was observed between embryo quality, mean number of cells, and fragmentation on day 3 among 25(OH)D/calcifediol-deficient, -insufficient, and -replete groups (*p* = 0.73 and *p* = 0.79) [18]; this was subsequently supported in a large retrospective cohort study (1883 women and 1720 men) by Jiang et al. in 2019, which also failed to demonstrate any correlation between serum 25(OH)D levels in women and men and embryo development (cleavage and blastocyst stage) after intracytoplasmic sperm injection (ICSI)/IVF [19].

It is interesting to note that calcitriol, the active form of the VDES, and progesterone have some remarkable structural analogies. For this reason, many reviews have compared the effects produced by the VDES and progesterone on the implantation process and pregnancy. A possible co-action effect has been described, defining the VDES as a steroid hormone system with progesterone-like activity. Calcifediol helps the endometrium to be receptive, supporting implantation and the course of pregnancy [20]. The availability of 25(OH)D/calcifediol appears to enhance the implantation process by inducing an immunological response in the intrauterine environment [5,18,21]. Moreover, it has been postulated that 25(OH)D/calcifediol may influence endometrial receptivity by increasing the expression levels of HOXA10, a transcription factor, in endometrial stromal cells [22] and the secretion of progesterone by the granulosa cells of the ovary, thus potentially providing a better endometrial environment [4].

The correlations between 25(OH)D/calcifediol serum levels and their influence on IVF outcomes have been extensively studied and analyzed in several meta-analyses and systematic reviews, with controversial results being reported [23,24,25]. Some systematic reviews have demonstrated that 25(OH)D/calcifediol insufficiency has an adverse effect on implantation rates, clinical pregnancy, and ongoing live birth after IVF/ICSI cycles [5,10,26], whereas others have shown a beneficial effect of 25(OH)D/calcifediol in replete patients. However, some limitations were encountered such as ethnic heterogeneity of the study population, small sample sizes, different study designs, different ovarian stimulation protocols, and further relevant confounders which were not considered in the analysis. The varying results reported among different studies could be explained by the use of different assays to measure 25(OH)D/calcifediol serum levels. Most laboratories rely on automated immunoassays to measure 25(OH)D/calcifediol, but these have some shortcomings related to intrinsic analytic issues and demonstrated fluctuating performance [27]. Recently, liquid chromatography–mass spectrometry (LC-MS/MS) has been proposed as the gold standard to achieve more precise 25(OH)D/calcifediol measurement, but its use requires complex, expensive equipment and well-trained staff [28]. Further studies are needed to accurately assess 25(OH)D/calcifediol serum levels.

One of the first groups to assess whether 25(OH)D/calcifediol serum levels could predict clinical pregnancy after IVF was Ozkan et al. in 2010 [12]. They reported that women with lower 25(OH)D/calcifediol levels in serum and follicular fluid are less likely to achieve a clinical pregnancy after IVF than 25(OH)D/calcifediol-sufficient patients (*p* = 0.041). Each ng/mL increase in 25(OH)D/calcifediol in follicular fluid led to a 6% increase in the chance of clinical pregnancy (*p* = 0.01) [12]. Even in spontaneous pregnancies, Jukic et al. reported that women with 25(OH)D/calcifediol serum levels < 20 ng/mL had a 45% reduction in fecundability [OR 0.55, CI (0.23, 1.32)] versus an increase of 35% in women with 25(OH)D/calcifediol serum levels > 50 ng/mL [OR 1.35, CI (0.95, 1.91)], suggesting a role of 25(OH)D/calcifediol in the mechanism of conception [29]. A retrospective study by Rudick et al. confirmed the beneficial effect of 25(OH)D/calcifediol on IVF pregnancies, as they found a significant correlation between 25(OH)D/calcifediol status and pregnancy rates after stratifying patients according to ethnic origin (*p* < 0.01) [18]. Moreover 25(OH)D/calcifediol was associated neither with oocyte yield nor with embryo parameters on day 3, suggesting a possible positive effect of 25(OH)D/calcifediol on IVF pregnancy only through the endometrium.

To distinguish whether the effects of 25(OH)D/calcifediol are mediated through the endometrium or oocyte quality, several studies investigated the effect of 25(OH)D/calcifediol on implantation by including only oocyte donation cycles; however, the results are controversial. Fabris et al. concluded that endometrial receptivity does not seem to be impaired by 25(OH)D/calcifediol status in recipients of donated oocytes, with comparable implantation, pregnancy, and ongoing pregnancy rates among 25(OH)D/calcifediol-replete (61%, 70%, and 55.9%, respectively), -deficient (63.4%, 69.9%, and 52.7%, respectively), and -insufficient patients (65.2%, 73.9%, and 60.7%, respectively) [30]. Contrary to this finding, Rudick et al. reported that the effect of 25(OH)D/calcifediol is mediated through the endometrium, since lower clinical pregnancy rates were observed when lower 25(OH)D levels (<20 ng/mL) were compared to normal 25(OH)D levels (37% vs. 78%; *p* = 0.004) in recipients of donated oocytes [31].

In line with previous findings, a recent small study demonstrated that administration of vitamin D supplements 6 weeks before the day of oocyte retrieval did not seem to improve fertilization rate, oocyte maturity, or oocyte/embryo quality, but enhanced endometrial quality and pregnancy rates [32]. Polyzos et al. demonstrated that, among patients undergoing single embryo transfer at blastocyst stage (day 5) without being screened genetically for aneuploidies, 25(OH)D/calcifediol-deficient patients (<20 ng/mL) had significantly lower clinical pregnancy rates compared to 25(OH)D/calcifediol-nondeficient patients (>20 ng/mL) (*p* = 0.015) [33]. Moreover, no clear effect of 25(OH)D/calcifediol status was reported on ovarian response to stimulation [33]. Farzadi et al. demonstrated that 25(OH)D/calcifediol can independently improve implantation rate and IVF outcome without affecting the number and quality of oocytes [34]. Higher clinical pregnancy rates were also reported by Paffoni et al. in 25(OH)D/calcifediol-replete patients compared with 25(OH)D/calcifediol-deficient/insufficient patients (31% vs. 20%, respectively; *p* = 0.02) [35].

Conversely, other groups were unable to document a benefit of 25(OH)D/calcifediol for IVF outcomes such as clinical pregnancy [23,24,25]. As previously described, a significant correlation has been reported between levels of 25(OH)D/calcifediol in follicular fluid and serum (r = 0.767, *p* = 0.001) [23]. Among mainly 25(OH)D-deficient patients, Aleyasin et al. reported no differences in median 25(OH)D/calcifediol levels in follicular fluid among pregnant versus nonpregnant women (9.19 ng/mL vs. 10.34 ng/mL, *p* = 0.433) [23]. In line with the previous findings, Firouzabadi et al. failed to find a correlation between pregnancy rate and serum 25(OH)D/calcifediol level (*p* = 0.094) or follicular fluid 25(OH)D/calcifediol level (*p* = 0.170), suggesting no association between 25(OH)D/calcifediol and success of IVF treatments [24]. Furthermore, Franasiak et al. did not find any association among the three groups of 25(OH)D/calcifediol levels analyzed (insufficient, deficient, and replete) and IVF outcome [25], when evaluating the influence of serum 25(OH)D levels and pregnancy outcome after the transfer of only euploid blastocyst(s). In agreement with the previous studies, there is no relation between serum 25(OH)D/calcifediol and the probability of clinical pregnancy or live birth [18,36].

Based on the available literature, it is still unknown whether 25(OH)D/calcifediol serum/follicular levels should be considered a prognostic factor for IVF success. It is possible that the optimal 25(OH)D/calcifediol level for reproductive success is still unknown and 25(OH)D/calcifediol serum/follicular fluid cut-off values need to be elucidated.

### 2.2. Anti-Müllerian Hormone (AMH) and Calcifediol Status

There is currently a debate as to whether 25(OH)D/calcifediol might influence ovarian reserves, specifically anti-Müllerian hormone (AMH) levels. It has been suggested that 25(OH)D/calcifediol might be a regulator of AMH production, as women with 25(OH)D/calcifediol serum levels < 30 ng/mL in follicular fluid have an increase in AMH receptor II mRNA expression in the granulosa cells of small follicles, suggesting an important role for 25(OH)D/calcifediol in AMH gene expression and signaling [37]. In another study by the same group, a positive correlation between serum 25(OH)D/calcifediol serum levels and AMH levels was found in late reproductive age (>40 years old) (regression slope = +0.011; *p* = 0.028). However, in women aged <35 years, and after adjustment for covariables, an insignificant correlation between 25(OH)D/calcifediol and AMH was observed (r^2^ = −0.0086; *p* = 0.054) [38]. Focusing on healthy young donors, Fabris et al. failed to demonstrate any correlation between 25(OH)D/calcifediol and AMH levels (r^2^ = 0.059) [30]. However, in women trying to conceive spontaneously, 25(OH)D/calcifediol levels were not found to correlate with AMH values, but there was a tendency for insufficient 25(OH)D/calcifediol (<30 ng/mL) to be associated with low AMH (<0.7 ng/mL) [OR 1.8, CI (0.9–4)] [39]. In a mainly Caucasian non-25(OH)D/calcifediol-deficient (69.3% ≥ 20 ng/mL) population, Drakopoulus et al. found no correlation between 25(OH)D/calcifediol and AMH [40]. Another cross-sectional study including infertile women with a high prevalence of diminished ovarian reserve confirmed a lack of association between 25(OH)D/calcifediol serum levels (<20 ng/mL vs. ≥20 ng/mL) and AMH levels (0.8 ± 3.0 ng/mL vs. 0.5 ± 1.6 ng/mL; *p* = 0.1761, respectively) after adjustment for age, BMI, and seasonal fluctuations [41]. Interestingly, after controlling for seasonal fluctuations, a negative linear correlation was found between AMH levels and 25(OH)D/calcifediol levels only up to approximately 30 ng/mL (*p* = 0.06). Beyond this value, there was no statistically significant relationship (*p* = 0.50) [42].

Further studies controlling for different cofounders and in larger groups of patients are needed to confirm the association between AMH and 25(OH)D/calcifediol.

### 2.3. Polycystic Ovary Syndrome and Calcifediol

Polycystic ovary syndrome (PCOs) is a common endocrinopathy which is characterized by irregular menstrual cycles, anovulatory infertility, excess androgens, insulin resistance, and often obesity [43]. Around 67% to 85% of women diagnosed with PCOs are known to be 25(OH)D/calcifediol-deficient [44]. Supporting the hypothesis that 25(OH)D/calcifediol is a negative regulator of AMH, there have been some interventional studies that evaluated the relationship between vitamin D supplementation and AMH according to the woman’s ovulatory status. Some authors demonstrated that serum AMH was significantly decreased following supplementation in women with PCOs (*p* < 0.001), while no change was observed in non-PCOs patients (*p* = 0.003) [45]. The same effect was observed among a 25(OH)D/calcifediol-deficient population when administering vitamin D supplements to only patients with PCOs compared with those receiving placebo (*p* = 0.02).

Undoubtedly, the mechanism of action of the VDES in PCOs and ovulation is complex and requires additional study.

### 2.4. The VDES and Uterine Fibroids

Uterine fibroids (UF; myomas or leiomyomas) are benign tumors of smooth muscle tissue in the uterus. They are the most common benign tumors in women of reproductive age (30–40%), with a lower incidence once menopause commences [46]. Around 50% of women presenting with UF experience some symptoms that might negatively impact their quality of life, including reduced fertility or higher miscarriage rates [47]. However, it is difficult to define an association between infertility and UF due to the heterogeneity of fibroids regarding location, size, and number, as well as the different prevalence rates observed among different patient populations [48].

In the past few years, several studies have investigated the potential biological effect of the VDES and its metabolites on the development of UFs. Basic research using in vitro and in vivo animal models demonstrated that the VDES suppresses cell proliferation and cell growth, causing a reduction in UF [49]. It has also been suggested that the VDES acts as a suppressor of transforming growth factor beta (TGF-b) which is involved in the development and progression of UFs [50].

A negative correlation between serum 25(OH)D/calcifediol serum levels and the presence of UFs has been observed. Sufficient 25(OH)D/calcifediol serum levels were associated with a 32% reduced risk of UFs compared to those with 25(OH)D/calcifediol insufficiency [OR = 0.68, CI (0.48–0.96)], regardless of ethnicity [51]. A similar prevalence of UFs among women with low serum 25(OH)D/calcifediol levels has been described by others [52,53]. Only one study initially found no association between insufficient 25(OH)D/calcifediol serum levels and the prevalence of UFs in a large US-based population [54]. However, the major limitation of that study is that UF diagnosis was only based on patient certainty that no previous UF was diagnosed.

While there is clear evidence that 25(OH)D/calcifediol causes molecular alterations in UF, there are controversial results regarding its clinical use as a potential therapy for UF management.

Vitamin D supplementation after 12 months restored 25(OH)D/calcifediol serum levels in women with hypovitaminosis D and reduced the growth of UFs, suggesting vitamin D supplementation as an effective therapeutic strategy to avoid surgical intervention for small fibroids (<5 cm in diameter) [52]. Even with a shorter period (10 weeks) of vitamin D supplementation, 25(OH)D/calcifediol serum levels were significantly higher in 25(OH)D/calcifediol-deficient patients than in the placebo group (36.08 ng/mL vs. 16.25 ng/mL; *p* < 0.001) and UFs decreased significantly in size [55]. However, in a recent RCT, vitamin D supplementation was not associated with a statistically significant reduction in the volume of UFs, but it did prevent their further growth [56]. To overcome the major limitations of previously published studies, in which only small numbers of subjects were included, an ongoing open-label RCT including more than 2000 Chinese individuals is currently evaluating the efficacy of vitamin D supplementation in reducing the incidence of UFs in women of reproductive age [57].

Despite the lack of a clear consensus, vitamin D supplementation or calcifediol treatment [6] could be a potential inexpensive treatment for the prevention of further UF growth and the treatment of UFs.

### 2.5. Vitamin D Supplementation and IVF

There is currently no recommendation for 25(OH)D/calcifediol testing prior to an IVF cycle. However, the question remains open: does vitamin D supplementation improve the outcomes of 25(OH)D/calcifediol-nonreplete women undergoing IVF? After analyzing costs and benefits, Pacis et al. determined that the approach of testing for 25(OH)D/calcifediol deficiency and giving supplementation when needed before starting an IVF cycle may be beneficial, with a reported increase of 3% in ongoing pregnancy rates (from 35% to 38%) [58].

However, there have been some recent interventional studies aiming to test the potential benefit of vitamin D supplementation in improving clinical pregnancy rates among women undergoing IVF. In a randomized, double-blinded, placebo-controlled study (SUNDRO study), women with 25(OH)D/calcifediol serum levels <30 ng/mL were randomized to receive 600,000 IU of vitamin D or placebo 2–12 weeks before oocyte retrieval. Interestingly, clinical pregnancy rates were not significantly different between the two groups (37% and 40%, respectively; *p* = 0.37) [59].

Based on existing studies, it remains unclear whether supplementation with vitamin D and monitoring 25(OH)D/calcifediol serum levels to ensure repletion can directly improve IVF outcome or not; however, some of the evidence suggests that it may have a positive impact, and it is a cost-effective and simple treatment for health-related issues that may be linked to fertility health markers. Due to its special pharmacological characteristics [6], the use of calcifediol treatment for the correction of 25(OH)D/calcifediol deficiency should be investigated further.

## 3. VDES and Pregnancy Outcomes

The influence of the VDES on pregnancy outcomes is the subject of similar controversies to its relationship with human fertility. Over an extended period, various observational studies have managed to show an existing relationship between low 25(OH)D/calcifediol serum levels before and at the beginning of pregnancy and poor perinatal outcomes. However, it is necessary to recognize that the heterogeneity of these studies impacts an objective analysis of the accumulated scientific evidence. This heterogeneity refers to the number of individuals recruited in some studies, the different evaluation time points considered by the different authors, the variety of results analyzed, and sometimes even differences in the methodology used for the analytical quantification of 25(OH)D/calcifediol [60]. It is also worth noting that there is growing evidence that receiving vitamin D supplements during pregnancy might reduce the risk of obstetric complications such as gestational diabetes (GD) [61] and preeclampsia [62], and may improve newborn birth weight [63,64].

Several observational studies and meta-analyses have shown that pregnancy is a crucial period in which 25(OH)D/calcifediol deficiency may affect maternal and neonatal outcomes [65]. Positive associations between 25(OH)D/calcifediol status and adverse pregnancy outcomes such as preeclampsia, GD, preterm birth, and small size for gestational age (SGA) have been reported, suggesting that hypovitaminosis D influences the risk of adverse maternal and neonatal outcomes [66]. In fact, RCTs indicate that vitamin D supplementation during pregnancy optimizes maternal and neonatal 25(OH)D/calcifediol status [67]. To date, the ideal dose for the treatment of 25(OH)D/calcifediol-deficient infertile patients or pregnant women has not been sufficiently clarified and most clinical studies have been carried out using cholecalciferol. Overall, calcifediol has been shown to be safe during this particular period, enabling its use in appropriate RCTs with good methodological designs and adequate patient numbers to definitively clarify the influence of the VDES both in infertility and during gestation [60]. Due to its special pharmacokinetic characteristics, the use of calcifediol seems adequate for correcting 25(OH)D/calcifediol deficiency in women of reproductive age, as well as those expressing the desire for conception and supplementation during pregnancy.

### 3.1. VDES and Birth Weight

It is well known that what happens from the earliest stages of embryonic development during pregnancy has important consequences for later susceptibility to different chronic diseases. In this way, various associations have been found between the weight of the newborn and its subsequent risk of experiencing cardiovascular disease in general, and even type 2 diabetes. It is extremely complex, however, to correlate the very different environmental influences, such as nutritional factors, that can produce a variety of neonatal phenotypes that could affect the risk of cardiovascular and other diseases during adult life, even in the absence of effects on the fetus’s own weight at birth. The VDES is essential for fetal and childhood skeletal development, and experimental animal studies support an active contribution of the VDES to organ development. Since the VDES is involved in a wide variety of physiological processes, including cell differentiation and skeletal development, its status during pregnancy can affect infant birth size [66].

Several studies have reported a positive association between maternal vitamin D levels in pregnancy and offspring birth weight, although results from both observational studies and RCTs are inconsistent. A recently published Cochrane review [68] suggested a significant effect of maternal 25(OH)D/calcifediol on birth weight. In a recent meta-analysis which compared serum levels of 25(OH)D/calcifediol and risk of low birth weight, the authors reported that maternal 25(OH)D/calcifediol deficiency led to an increased risk of low birth weight. The conclusion was that the evidence from these results indicates a consistent association between 25(OH)D/calcifediol deficiency during pregnancy and an increased risk of low birth weight [69].

In the subsequent literature, great interest has been generated by the relationship between vitamin D supplementation during pregnancy and results from a systematic review that sought to assess the effects of oral vitamin D supplementation during pregnancy on different anthropometric variables of the fetus, such as birth weight, fetal length, head circumference, low birth weight (LBW), and small SGA at the time of delivery. This meta-analysis and systematic review finally confirmed that the VDES is essential for fetal growth and development throughout pregnancy, with well-established effects on the size of the fetus eventually determined. It was the first meta-analysis to demonstrate a significant positive effect of maternal vitamin D supplementation on the risk of SGA [70].

### 3.2. The VDES and Preeclampsia

Preeclampsia is a multisystem disorder that typically affects 2–5% of pregnant women and is one of the leading causes of maternal and perinatal morbidity and mortality, especially when the condition is of early onset. Its etiology remains unclear to date. Preeclampsia is best described as a pregnancy-specific syndrome that can affect virtually every organ system. In addition, it heralds a higher incidence of cardiovascular disease later in life. Although preeclampsia is much more than simply gestational hypertension with proteinuria, the appearance of proteinuria remains an important diagnostic criterion. Thus, proteinuria is an objective marker and reflects the system-wide endothelial leak that characterizes preeclampsia syndrome.

It is known that 25(OH)D/calcifediol deficiency is common during pregnancy [60,65]. In recent years, numerous studies have placed the VDES in the spotlight, as it has been hypothesized to act as a protective factor for preeclampsia due to its important role in the maintenance of immune homeostasis. VDES acts to control regulatory T cells, which ultimately prevents placental vasoconstriction, whilst also downregulating proinflammatory cytokines and avoiding an excessive and perpetuated proinflammatory environment. Other mechanisms proposed for this vitamin include regulation of smooth muscle cell proliferation and angiogenesis, and reduction of cholesterol uptake by arterial wall cells [64].

Several studies have tried to address what association exists between maternal 25(OH)D/calcifediol serum levels and preeclampsia. A recent systematic review and meta-analysis aimed to determine if maternal 25(OH)D/calcifediol insufficiency and/or deficiency during pregnancy was associated with the prevalence of preeclampsia and prematurity. This meta-analysis succinctly concluded that higher 25(OH)D/calcifediol concentrations during pregnancy could be associated with a decreased risk of preeclampsia and prematurity, but that statistical significance of any associations depended on the study design used [64]. Therefore, well-designed clinical trials with vitamin D supplementation or calcifediol [6] treatment are needed to better define these associations.

A recent study in a Chinese population that analyzed a cohort of more than 13,000 pregnant women showed that maternal 25(OH)D/calcifediol deficiency analyzed between weeks 23 and 28 of pregnancy had a strong association with an increased risk of experiencing severe forms of preeclampsia, even after adjusting for relevant confounding factors from a statistical point of view [71]. Similarly, another study, published in 2017, showed that supplemental administration of vitamin D during pregnancy could enhance medical treatment with nifedipine for preeclampsia, reducing the time needed to control blood pressure and lengthening the time before the next hypertensive crisis coma, probably through an immunomodulatory mechanism of the VDES [72].

With a higher level of scientific evidence, a recent 2019 systematic review and meta-analysis of RCTs evaluated 27 studies, with a total of 59 therapeutic arms including 2487 pregnant women treated with vitamin D who were compared against a total of 2290 that were attributed to the control arm [73]. Vitamin D treatment in pregnancy was associated with a reduced risk of preeclampsia. On the other hand, when vitamin D treatment was started around 20 weeks of pregnancy, the probability was slightly lower; the effect achieved was independent of the cessation of treatment, the type of intervention (either vitamin D alone or associated with calcium), and the study design. The authors concluded that increasing the dose of vitamin D during gestational treatment was associated with a reduced incidence of preeclampsia.

In a systematic review and meta-analysis published in 2017 by Serrano et al. [74], the authors reported an inverse relationship between the levels of 25(OH)D/calcifediol serum levels and the risk of developing preeclampsia, concluding that this association suggests that the higher the levels of 25(OH)D/calcifediol, the lower the probability of developing preeclampsia, despite the heterogeneity of the global measurement of these types of analytical results [74]. Overall, the evidence analyzed here allows us to suggest that 25(OH)D/calcifediol deficiency can seriously affect the risk of developing preeclampsia during pregnancy and, consequently, vitamin D supplementation or calcifediol treatment could be an adequate gestational intervention strategy to prevent preeclampsia as a major complication of pregnancy.

### 3.3. The VDES and Gestational Diabetes

Gestational diabetes (GD) is a common outcome of pregnancy, defined as any grade of glucose intolerance diagnosed during pregnancy, mostly after 24 weeks of gestation. GD is associated with an increased risk of short- and long-term consequences for the health of the mother and the fetus. The global prevalence of total hyperglycemia in pregnancy was estimated to have been 16.9%, or 21.4 million live births (women aged 20–49 years) in 2013 [75]. Knowing that both the insufficiency and deficit of maternal 25(OH)D directly influence the results obtained from pregnancy, the possible influence of 25(OH)D on the development of GD has been evaluated in the scientific literature [76].

In this sense, a recent meta-analysis, published in 2020, tried to establish the relationships between the appearance of GD and maternal levels of 25(OH)D/calcifediol. The authors verified that the vitamin D levels of people with GD were much lower than those of healthy women and concluded, in the other direction, that vitamin D deficiency was associated with a high risk of developing GD. Finally, in light of these results, it was concluded that 25(OH)D/calcifediol is very closely associated with the risk of developing GD [77].

Additionally, another meta-analysis and systematic review of prospective scientific studies specifically evaluated the association between the risk of developing gestational diabetes and maternal blood levels of vitamin D. It was confirmed that women with 25(OH)D/calcifediol deficiency had up to a 26% higher risk of developing GD than those with normal 25(OH)D/calcifediol serum levels, and a positive and significant association was observed between the combined insufficiency and deficiency of 25(OH)D/calcifediol and the risk of developing GD [78]. Dose–response analysis showed a significant U-shaped nonlinear association between serum 25(OH)D/calcifediol concentration and the risk of developing GD (*p* < 0.001), such that those with serum 25(OH)D/calcifediol concentrations between 16 and 36 ng/mL had a significantly reduced risk of GD.

Finally, a 2019 systematic review evaluated the advisability of administering vitamin D supplements to women with GD to try to adjust their glycemic control [79]. Results of fasting blood glucose, glycated hemoglobin level, and serum insulin were evaluated. The overall conclusion was that vitamin D supplementation was associated with a marked decrease in fasting blood glucose concentration of glycosylated hemoglobin and insulin compared to the control group. Once again, this recent systematic review showed sufficient evidence that vitamin D supplementation has the potential to promote adequate blood glycemic control in women with GD.

## 4. Conclusions

The VDES, in which cholecalciferol or vitamin D is the nutrient and calcifediol the prohormone and indispensable substrate for the synthesis of the active hormone calcitriol, participates in many biological functions. The presence of VDRs in several sites related to fertility unequivocally implies a necessary role in the development of both male and female fertility. Lower 25(OH)D/calcifediol levels in follicular fluid are associated with lower embryo quality, fertilization, implantation, and clinical pregnancy rates. To date, we cannot definitively conclude the relationship between 25(OH)D/calcifediol levels and the endometrium in terms of its influence on embryo implantation, because the results of different studies have been contradictory. Similarly, and with regard to the assessment of ovarian reserve, it is necessary to control the different confounding factors in a study with a large number of patients to confirm or rule out the association between the VDES and AMH. It has not been possible to clearly determine the mechanisms of action that 25(OH)D/calcifediol exerts on ovulation in women with PCOS, so additional studies are necessary. Despite the fact that there is no general consensus, therapeutic supplementation with vitamin D, or calcifediol treatment, could be a potentially employable form of medical treatment for preventing the growth of UFs. Vitamin D supplementation and calcifediol treatment may have a positive impact, and it is a cost-effective and simple treatment that may be linked to fertility health markers such as pregnancy rate upon IVF. Despite their heterogeneity, several observational studies and meta-analyses have shown that pregnancy is a crucial period in which 25(OH)D/calcifediol deficiency can affect maternal and neonatal outcomes. Several studies have reported a positive association between maternal 25(OH)D/calcifediol levels during pregnancy and offspring birth weight, although the results of all studies remain inconsistent. Low levels of 25(OH)D/calcifediol are related to the risk of developing preeclampsia, and treatment during pregnancy could be an appropriate gestational intervention strategy to prevent preeclampsia as a major complication of pregnancy. Various studies have provided sufficient evidence that adequate 25(OH)D/calcifediol ensured with vitamin D supplementation has the potential to promote adequate control of blood glucose in women with GD. The heterogeneity of the studies carried out so far, as well as the inconsistent results in all cases, make further research studies necessary to elucidate the true role of calcifediol treatment in improving human fertility as well as in gestational outcomes in terms of maternal–fetal health. To date, there is insufficient scientific evidence for vitamin D supplementation/treatment during pregnancy, so well-designed, prospective clinical intervention trials with sufficient patient numbers are necessary. Cholecalciferol treatment has generally been used more widely, both in infertility and during pregnancy, due, among other factors, to its much longer presence in daily care guidelines, although calcifediol has been shown to be sufficiently safe to be evaluated in RCTs.

## Figures and Tables

**Table 1 nutrients-14-01820-t001:** Characteristics of the studies regarding 25(OH)D and embryo development/implantation/clinical pregnancy/live birth rate after IVF/CSI included in the review.

Author(s), Year	Study Design	Participants (n) and Main Inclusion Criteria	Intervention	Source of Sample	Clinical Outcome Measures	25(OH)D STATUS	25(OH)D Measured	Conclusion
Muyayalo et al., 2021 [14]	Prospective cohort study	132 IVF patients	130 fresh ET on day 3 or FET at blastocyst stage	Pooled follicular fluid	Fertilization embryo quality, IR and CPR	Deficient < 20 ng/mL; Insufficient 20–29 ng/mL; Replete ≥ 30 ng/mL	25(OH)D	25(OH)D levels in FF but not in serum were associated with fertilization, embryo quality, IR and CPR
Ciepiela et al., 2018 [15]	Prospective cohort study	198 IVF patients	88 fresh SET on day 3 and 18 ETs on day 5	Pooled follicular fluid and serum	Embryo quality	Deficient < 20 ng/mL; Sufficient ≥ 20 ng/mL	25(OH)D	25(OH)D levels in FF correlates negatively with fertilization and embryo development.
Anifandis et al., 2010 [16]	Prospective cohort study	101 IVF patients	86 fresh ET on day 3	Pooled follicular fluid	Embryo quality and IVF outcomes	Deficient < 20ng/mL; Insufficient 20–29 ng/mL; Replete ≥ 30 ng/mL	25(OH)D and Glucose levels	Increased 25(OH)D levels in combination with decreased glucose levels have a negative impact on embryo quality and therefore on IVF outcome
Arnanz et al., 2021 [17]	Prospective observational study	37 IVF patients	114 biopsied blastocysts	Individual follicular fluid and serum	Blastocyst ploidy status	Deficient < 20 ng/mL; Non-deficient ≥ 20 ng/mL	25(OH)D, bioavailable 25(OH)D, free 25(OH)D and % free 25(OH)D	25(OH)D non-deficient patients have a significantly higher probability of obtaining a euploid blastocyst compared to VitD deficient patients
Rudick et al., 2012 [18]	Retrospective cohort study	188 infertile women undergoing IVF treatment	Fresh ET on day 3 and day 5	Serum	Embryo quality mean number of cells, fragmentation on day 3 and CPR	Deficient < 20 ng/mL; Insufficient 20–29 ng/mL; Replete ≥ 30 ng/mL	25(OH)D	VitD deficiency is associated with lower pregnancy rates in non-hispanic whites. VitD deficiency was not associated with IVF outcomes
Jiang L. et al., 2019 [19]	Retrospective cohort study	1883 women and 1720 men undergoing IVF treatment	Fresh ET on day 3	Serum in women and men	Embryo development at cleavage and blastocyst stage. IR, CPR, miscarriage rate and LBR	Not specified	25(OH)D	No correlation between serum 25(OH)D levels in women and men and embryo development (cleavage and blastocyst stage) and clinical outcomes
Ozkan et al., 2010 [12]	Prospective cohort study	84 IVF patients	Fresh ET on day 3	Serum and follicular fluid	CPR	Deficient < 20 ng/mL; Insufficient 20–29 ng/mL; Replete ≥ 30 ng/mL	25(OH)D	High 25(OH)D in FF and serum levels were related to higher CPR
Fabris et al., 2014 [30]	Retrospective study	267 recipients of donated oocytes	Fresh ET on day 3	Serum	IR, CPR and OPR	Deficient < 20 ng/mL; Insufficient 20–29 ng/mL; Replete ≥ 30 ng/mL	25(OH)D and bioavailable 25(OH)D	No significant correlation between 25(OH)D levels and CPR in recipients of donated oocytes
Rudick et al., 2014 [31]	Retrospective cohort study	99 recipients of donated oocytes	Fresh ET on day or 5	Serum	CPR in donor-recipient IVF cycles	Deficient < 20 ng/mL; Insufficient 20–30 ng/mL; Replete > 30 ng/mL	25(OH)D	25(OH)D < 30 ng/mL levels in recipients of donated oocytes showed lower PR
Abedi et al., 2019 [32]	Double-blind clinical trial	108 IVF patients randomly allocated: VitD supplements 6 weeks before oocyte retrieval (n = 54) or placebo as a control group (n = 54)	VitD supplementation (42 participants) and placebo (43 participants)	Serum	Number of oocytes retrieved, oocyte maturity, fertilization rate, rate of embryo quality, endometrial quality and CPR	Deficient < 30 ng/mL	25(OH)D	25(OH)D supplementation is effective in improving the clinical outcome of ICSI
Polyzos et al., 2014 [33]	Retrospective cohort study	508 IVF patients undergoing SET on day 5	368 IVF patients undergoing SET on day 5	Serum	Ovarian response to stimulation and CPR	Deficient < 20 ng/mL; Insufficient 20–30 ng/mL; Replete > 30 ng/mL	25(OH)D	Low 25(OH)D levels were related to lower CPR and LBR
Farzadi et al., 2015 [34]	Prospective observational study	80 IVF patients	80 fresh ET on day 3	Serum and pooled follicular fluid	Number and quality of oocytes and IR	Not specified	25(OH)D	25(OH)D levels don’t affect the number and quality of oocytes but higher 25(OH)D levels improve IR and IVF outcome
Aleyasin et al., 2010 [23]	Prospective cohort study	82 IVF patients	77 fresh ET on day 3	Serum and pooled follicular fluid	CPR	Deficient < 20 ng/mL; Insufficient 20–30 ng/mL; Replete > 30 ng/mL	25(OH)D	No significant CPR among different 25(OH)D levels
Firouzabadi et al., 2013 [24]	Prospective observational study	180 IVF patients	495 ETs	Serum and pooled follicular fluid	PR	Deficient VitD < 10 ng/mL; Insufficient VitD 10–29 ng/mL; Sufficient VitD 30–100 ng/mL	25(OH)D	No correlation between 25(OH)D levels in serum and FF and PR
Franasiak et al., 2015 [25]	Retrospective cohort study	529 IVF patients that went through a PGT-A cycle	517 IVF patients that went through single euploid frozen embryo transfer on day 6	Serum	PR	Deficient VitD < 20 ng/mL; Insufficient VitD 20–30 ng/mL; Replete > 30 ng/mL	25(OH)D	No correlation between 25(OH)D levels and CPR in women undergoing euploid embryo transfer
Abadia et al., 2016 [36]	Prospective cohort study	100 IVF patients	168 initiated IVF cycles, 141 IVF cycles with oocyte retrieval	Serum	CPR or LBR	Deficient VitD 13.5–30 ng/mL; Sufficient VitD 30.5–62.3 ng/mL	25(OH)D	25(OH)D levels were unrelated to CPR or LBR after IVF
Paffoni et al., 2014 [35]	Prospective cross-sectional study	480 IVF patients	335 fresh ETs. 154 patients were VitD deficient vs. 181 VitD insufficient	Serum	CPR, IR	Deficient VitD < 20 ng/mL; Insufficient VitD 20–29 ng/mL	25(OH)D	Higher 25(OH)D levels were associated with higher CPR and IR

Abbreviations. 25(OH)D: 25-hydroxyvitamin D. IVF: In vitro fertilization, ICSI: Intracytoplasmic Sperm Injection, FF: Follicular fluid, ET: Embryo transfer, FET: Frozen Embryo Transfer, SET: Single Embryo Transfer. IR: Implantation rate, CPR: Clinical Pregnancy Rate, LBR: Live Birth Rate.

## Data Availability

Not applicable.

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
