# Peer review of "Calcifediol (25OHD) Deficiency and Its Treatment in Women’s Health and Fertility"

_nutrients, 2022, doi:10.3390/nu14091820_

Round 1

Reviewer 1 Report

The authors conducted a interesting work about the Calcifediol (25OHD) deficiency and its treatment in women’s 2 health and fertility. Whole work is well writen and gives new evidence about the importance of vitamin D statusi in womens health. The review is carefully prepared and the conlusions are quite interesting. The results obtained are create a new nead of further studes in presented mather. Authors informe that large prospective clinical intervention studies are needed to clarify the associated benefit of this supplementation and the optimal dose to use in each situation.

In my opinion some parts of the text should be rewritten to better presentation and understand the significance of the presented problem and the complexity of the supplementation protocols that are used (ex. The Vitamin D Endocrine System and Infertility).

In addition Authors should show the problem of patients that are characterized by low reactivity to supplementation doses. Souch situtation is confirmed in numerous scientific studies and in case of womens health it may be importaint factor.

Author Response

All the authors sincerely appreciate the reviewer's comments, which undoubtedly help us to improve our professional activity and allow us to deepen our knowledge of all the mechanisms that 25 hydroxy vitamin D has for women's health.

Regarding the suggestions about improving the levels and the treatment of infertile patients, we make them our own and even plan to undertake these comments in the near future through a new manuscript specifically dedicated to the treatment of these situations.

We think that doing so at this time could lengthen the manuscript too extensively according to the initial intention and probably would not be acceptable to the editors.

We sincerely appreciate all the suggestions that will allow us to continue improving our professional activity.

Thank you sincerely.

Reviewer 2 Report

I read the manuscript "Calcifediol (25OHD) deficiency and its treatment in women’s health and fertility" with great interest.  It is a well written and scientificially solid based draft. 

Authors summarised the role of 25-hydroxyvitamin D (calcifediol) deficiency in common women diseases and infertility. They summarised the evidence on the field from the vitamin D endocrine system perpective and I believe they covered all the major topics in the area. 

I only have 3 minor comments/recommendations to improve the manuscript:

  1. Line 109-110, I was not able to see any table in the manuscript that I reviewed and I did not have and accsess to the supplementary file if there is one.  Please make sure that the table is available for reader.
  2. I would change the title pregnancy outcomes since you talk about both maternal and fetal outcomes in there. You could use VDES and pregnancy then subtitles as maternal outcomes and fetal outcomes. MAybe then you could include more fetal outcomes rather than only birth weight since there are many more. Also the same for maternal outcomes.
  3. I would also recommend authors to discuss bolus, high dose or daily dose vitamin D treatments and their different outcomes. That is a hot topic at the moment and there are many new publications on the area. BUt of course it is a recommendation so I would respect if the authors would want to leave it out. 

Author Response

All the authors sincerely appreciate all the comments that enhance our work and, above all, help us improve our professional activity.

Regarding the table that is missing, it is undoubtedly an unforgivable mistake and we are attaching it right now.

On the other hand, and with regard to the differentiated results of pregnancy or maternal and newborn, we intend to point out only some of all those published so as not to excessively lengthen the manuscript. On the other hand, the most controversial ones would have forced us to an extension difficult for the editors to accept.

In the same way, we did not include the issue of doses and the different ways of approaching each situation, because we think that it could even be the subject of a later publication, more extensive and focused exclusively on therapeutic aspects.

We appreciate again all your suggestions that do nothing but help us improve.

Thank you very much indeed!!!!

Reviewer 3 Report

In this manuscript authors reviewed the role of the VDES in human fertility evaluating the relationships between 25-hydroxyvitamin D (calcifediol) deficiency and in vitro fertilization, as well as aspects related to ovarian reserve and fertility, and commonly diagnosed endocrinopathies such as polycystic ovary disease. Moreover, authors briefly reviewed the relationships between calcifediol deficiency and uterine fibroids, as well as the role that treatment may have in improving human fertility. Finally, it has been screened the current literature regarding the consequences of calcifediol deficiency during pregnancy. 

The manuscript is well written and gives an overall status of the current studies regarding  Calcifediol (25OHD) deficiency in the complications described. However, some points should be improved. In particular:

  • "3.2. The VDES and preeclampsia": Definition of preeclampsia is too simplified. Authors should highlight the complexity of PE, in fact this pathology is also characterized by trophoblast immaturity (PMID: 32529396) and vascular dysfunction (PMID: 34831277). 
  • Line 400-408: Please insert references
  • A table for each chapter would help to better understand the studies described.
  • Authors should consider adding a schematic figure summarizing the role of VDES and pregnancies complications described.

Author Response

All the authors sincerely appreciate the interest with which the reviewer has read our manuscript and we deeply appreciate all the suggestions that have been made about it, because they do nothing but try to improve our already high interest in the subject of the role of the endocrine system of vitamin D and women's health.

We have made a table that summarizes a large part of the manuscript and the revision carried out, but that has not been included surely due to an unforgivable oversight that we correct at this very moment by attaching it to this comment.

We very sincerely and equally appreciate all the comments on the definition of preeclampsia, which, indeed, has been excessively simple in our manuscript. Notwithstanding the foregoing, our initial intention and the objective of this review was not so much to delve into the pathophysiology of the aforementioned condition (for which we greatly appreciate the references provided), as to express the relationship between the values of 25 hydroxyvitamin D and the increase in risk of appearance of the aforementioned complication of pregnancy.

We take note of the references provided for future approaches to the issue in a future manuscript.

Thnk you very much indeed!!!
